# Brown and Green Seaweed Antioxidant Properties and Effects on Blood Plasma Antioxidant Enzyme Activities, Hepatic Antioxidant Genes Expression, Blood Plasma Lipid Profile, and Meat Quality in Broiler Chickens

**DOI:** 10.3390/ani13101582

**Published:** 2023-05-09

**Authors:** Mohammad Naeem Azizi, Teck Chwen Loh, Hooi Ling Foo, Henny Akit, Wan Ibrahim Izuddin, Danladi Yohanna

**Affiliations:** 1Department of Animal Science, Faculty of Agriculture, Universiti Putra Malaysia, UPM Serdang, Seri Kembangan 43400, Selangor, Malaysia; naimazizi83@gmail.com (M.N.A.); henny@upm.edu.my (H.A.); wanahmadizuddin@gmail.com (W.I.I.); avyohanna@gmail.com (D.Y.); 2Department of Pre-Clinic, Faculty of Veterinary Science, Afghanistan National Agricultural Sciences and Technology University, ANASTU, Kandahar 3801, Afghanistan; 3Institute of Tropical Agriculture and Food Security, Universiti Putra Malaysia, UPM Serdang, Seri Kembangan 43400, Selangor, Malaysia; 4Department of Bioprocess Technology, Faculty of Biotechnology and Biomolecular Science, Universiti Putra Malaysia, UPM Serdang, Seri Kembangan 43400, Selangor, Malaysia; hlfoo@upm.edu.my; 5Institute of Bioscience, Universiti Putra Malaysia, UPM Serdang, Seri Kembangan 43400, Selangor, Malaysia

**Keywords:** antioxidant enzyme activity, antioxidant gene expression, broiler chickens, blood plasma, brown seaweed, green seaweed, meat composition, meat quality

## Abstract

**Simple Summary:**

It has been widely reported that seaweed has numerous bioactive molecules that have been examined for health and growth-promoting effects. The current study was conducted to analyze the effects of various levels (starting from 0.25% to 1.25%) of brown seaweed and green seaweed on blood plasma antioxidant enzyme activities, hepatic antioxidant genes expression, blood plasma lipid profile, breast meat quality, and chemical composition in broiler chickens. The result showed that different levels of brown seaweed and green seaweed had no significant effects on broiler blood plasma catalase, superoxide dismutase, and glutathione peroxidase enzyme activities. In contrast, the hepatic superoxide dismutase 1 gene mRNA expression was significantly higher for birds fed 0.50% and 0.75% brown seaweed. Meanwhile, the total cholesterol and high-density lipoprotein levels were higher in blood plasma for birds fed 0.75 and 1% brown seaweed compared to the negative and positive control groups. The findings showed that different levels of brown seaweed and green seaweed had significantly higher breast meat crude protein content. The current research findings are useful for further studies to investigate the mechanisms and components responsible for affecting the hepatic antioxidant gene expression and blood plasma profile.

**Abstract:**

The study was designed to analyze the effects of brown seaweed (BS) and green seaweed (GS) on blood plasma antioxidant enzyme activities, hepatic antioxidant genes expression, blood plasma lipid profile, breast meat quality, and chemical composition in broiler chickens. The dietary treatment groups contained basal diet [negative control (NC)], basal diet + vitamin E (100 mg/kg feed) [positive control (PC)], basal diet + 0.25, 0.50, 0.75, 1, and 1.25% BS and GS supplements separately. The findings showed that both BS and GS exhibited remarkable antioxidant activity. In contrast, the maximum antioxidant activity was recorded by BS (55.19%), which was significantly higher than the GS (25.74%). Results showed that various levels of BS and GS had no significant effects on broiler blood plasma catalase (*CAT*), superoxide dismutase (*SOD*), and glutathione peroxidase (*GPx*) enzyme activities. The hepatic superoxide dismutase 1 (*SOD1*) gene mRNA expression was significantly higher for birds fed 0.50% and 0.75% BS. Regarding the plasma lipid profile, the total cholesterol (TC) and high-density lipoprotein (HDL) levels were higher (*p* < 0.05) for birds fed 0.75 and 1% BS compared to the negative and positive control groups. The findings showed that different levels of BS and GS had significantly higher breast meat crude protein (CP) content.

## 1. Introduction

Seaweed is a composition of macroalgae of various sizes, colors, and compositions. It has no specifically marked stems, flowers, and leaves [1], and it absorbs light through a wide variety of pigments for photosynthesis. Chlorophylls, carotenoids (carotenes and xanthophylls), and phycobiliproteins are the main photosynthetic pigments in macroalgae [2]. The pigments in the seaweed can be divided into three main groups: brown seaweed (Phaeophyceae), green seaweed (Chlorophyceae), and red seaweed (Rhodophyceae) [3,4].

The poultry industry significantly contributes to agri-food value chains. Therefore, seaweed has the potential to be utilized in animal feeds as a rich source of essential nutrients. A study conducted by Azizi et al. [5] reported that brown seaweed (BS) and green seaweed (GS) contained 59.8 and 55.88% CP, 1.28 and 0.30% ether extract (EE), 5.78 and 5.19% crude fiber (CF), 29.19 and 34.68% carbohydrates, and 9.7 and 9.14% ash contents, respectively. Regarding the mineral contents, BS and GS contained 0.14 and 0.13% Ca, 0.18 and 0.14% Na, 2.96 and 2.20% K, 0.73 and 0.55% Mg (10.73 and 8.54 mg). An amount of 100 g^−1^ Zn (3.21 and 2.63 mg) was also observed. An amount of 100 g^−1^ Cu (14.67 and 11.73 mg) was also observed. An amount of 100 g^−1^ Fe and 13.34 and 11.14 mg. An amount of 100 g^−1^ Mn contents, respectively, were also observed. They [5] also reported the amino acids composition of seaweed and stated that BS and GS have 13.66 and 11.34 ng/mg lysine, 4.76 and 2.84 ng/mg leucine, 1.53 and 2.40 ng/mg phenylalanine, 24.24 and 8.41 ng/mg threonine, 9.41 and 8.32 ng/mg arginine, 4.18 and 25.41 ng/mg glycine, 4.05 and 2.57 ng/mg aspartic acid, 12.34 and 5.70 ng^1^/mg glutamic acid, 10.13 and 8.40 ng/mg serine, and 3.77 and 3.30 ng/mg alanine contents, respectively.

Furthermore, seaweed is a fascinating natural source of biologically active compounds used as medicinal components [6]. Among the other bioactive compounds in seaweed, polysaccharides have been proven to have various beneficial properties, such as antioxidant, anticoagulant, anti-inflammatory, antiviral activities, and anticarcinogenic [7].

Numerous free radical species can damage organic molecules, including carbohydrates, lipids, and proteins. It also damages DNA in different mechanisms, such as disrupting DNA duplication, interfering with DNA maintenance, breaking open the molecule, or altering the structure by reacting with the DNA bases [8]. Meanwhile, lipid oxidation is a major cause of food deterioration that may affect the color, flavor, texture, and nutritional value of poultry meat [9]. Therefore, natural antioxidants have been of high interest in recent years. Seaweed contains dietary antioxidants that can be used as medicinal and preventative agents [6,10,11].

Phenolic compounds of seaweed that have more than one hydroxyl group are considered effective primary antioxidants. The phenolic compounds’ antioxidant property is due to their capability to donate an H atom to free radicals and create relatively unreactive phenoxyl radicals [12]. Furthermore, sulphated polysaccharides, such as ulvan and fucoidan, also have substantial antioxidant activities [11]. The antioxidant mechanism of sulphated polysaccharides could be due to the hydrogen of polysaccharides, which combines with free radicals and forms a stable radical to cut off the radical chain reaction [13,14]. A study conducted by Airanthi et al. [15] determined that fucoxanthin and phenolics were responsible for the high antioxidant activities tested. The quest for alternative natural antioxidants has been studied because of the adverse effects of synthetic antioxidants on organisms [15,16]. The presence of additional antioxidant content, such as β-carotene, vitamin C, and α-tocopherol could contribute to brown, green, and red seaweed antioxidant efficacy. Natural tocopherol, ascorbic acid, and caffeic acid work as primary antioxidants, but they are typically less efficient than synthetic ones [12]. Polyphenolic compounds, sulfated polysaccharides, and carotenoids are the most significant antioxidants derived from different types of seaweed [12]. The current study was conducted to examine the effects of different levels of BS and GS on antioxidant status, blood plasma profile, and meat quality in broiler chickens.

## 2. Materials and Methods

### 2.1. Birds, Diets, and Experimental Design

This study was conducted in the Poultry Unit, Department of Animal Science, Universiti Putra Malaysia (UPM), following the guidelines approved by the Institutional Animal Care and Use Committee (IACUC) (UPM/IACUC/AUP-R093/2019).

A total of 504 day-old male broiler chickens (Cobb 500) were used for the experiment. The birds were labelled, weighed, and randomly allocated into twelve treatment groups. Each group had six replicates, with seven birds per replicate. The raising conditions followed commercial recommendations for Cobb 500. Birds were raised in a closed house with a penning cage system [120 × 120 cm (length × width)]. The house temperature was set at 32 ± 1 °C on day 1, and it was gradually reduced to around 24 ± 1 °C by the end of the trial. Birds were vaccinated against Newcastle and infectious bronchitis diseases (ND-IB) and infectious bursal disease (IBD) by eye drop at 7 and 21 days. The experimental dietary allocated groups were as follows: NC = negative control (basal diet), PC = positive control (basal diet + vitamin E, 100 mg/kg feed), BS 0.25 = basal diet + 0.25% BS, BS 0.50 = basal diet + 0.50% BS, BS 0.75 = basal diet + 0.75% BS, BS 1 = basal diet + 1% BS, BS 1.25 = basal diet + 1.25% BS, GS 0.25 = basal diet + 0.25% GS, GS 0.50 = basal diet + 0.50% GS, GS 0.75 = basal diet + 0.75% GS, GS 1 = basal diet + 1% GS, and GS 1.25 = basal diet + 1.25% GS. The starter period (Table 1) and finisher period (Table 2) diets were offered from days 0–21 and 22–42, respectively. The diets were formulated based on the broiler nutrient requirements (NRC, 1994) using the FeedLIVE software (FeedLIVE 1.60, Mueang Nonthaburi, Thailand).

### 2.2. Sample Collection

Dried seaweed samples were ground as a fine powder and used to determine antioxidant capacity. The seaweed was obtained from Promise Earth (M) Sdn. Bhd., Selangor, Malaysia.

At the end of the feeding trial, six chickens from each treatment were randomly selected and euthanized (cervical dislocation) for the sampling. Blood samples were collected into the blood collection tubes containing anticoagulant sodium heparin during the neck cutting and were stored in ice. The blood samples were then centrifuged at 3500× *g* for 15 min at 4 °C to harvest the plasma. The plasma was kept for enzymes activities determination and lipid profile analysis. Parts of the liver were collected, immediately frozen in liquid nitrogen, and stored at −80 °C for the gene expression analysis. In addition, approximately 30–40 g of fresh meat samples from the pectoralis major muscle were collected and immediately stored at 4 °C for meat quality traits and meat proximate composition determination.

### 2.3. Antioxidant Property of Brown and Green Seaweed

#### 2.3.1. Preparation of Seaweed Extract

The seaweed methanolic extract was prepared, as described by [17], with a minor modification. A 60 mg dried seaweed sample was weighed in a 1.5 mL tube, and 0.5 mL of methanol was added. The sample was sonicated for 20 min at 30 °C and centrifuged at 16,000× *g* for 5 min. The supernatants were collected in separated 1.5 mL tubes, and the residue was re-extracted two times under the same conditions.

#### 2.3.2. Antioxidant Activity Assay

The 2,2-Diphenyl-1-picryl-hydrazyl (DPPH) assay was conducted to determine the antioxidant activity of seaweed. The assay was performed according to the 96-well plate scheme, as described by [17]. Briefly, 10 µL from the early prepared sample extract was arranged in the 96-well plate, and 100 µL of DPPH solution (761 µM DPPH in 80% methanol) was added to each well. The mixture was incubated for 2 h in a dark chamber at room temperature. The absorbance was measured at 515 nm using a microplate spectrophotometer (Thermo Scientific™ Multiskan™ GO Microplate Spectrophotometer, Waltham, MA, USA). The equation below was used to calculate the percentage of DPPH free radical scavenging activity [18].
(1)$$\mathrm{DPPH}\;\mathrm{scavenging}\;\%=\left[\;\frac{\left(A0-\;\mathrm{As}\right)}{A0}\right]\times 100\;$$
where, A0 = absorbance of the DPPH alone, AS = absorbance of the sample.

### 2.4. Plasma Antioxidant Enzymes Activities

The plasma *CAT*, *SOD*, and *GPx* enzyme activities were measured using commercial kits, based on the quantitative colorimetric determination method.

#### 2.4.1. Catalase Activity

The *CAT* activity was measured using the EnzyChrom^TM^ catalase assay kit (ECAT-100, BioAssay Systems, Hayward, CA, USA). The test depends on the degradation of H_2_O_2_ using redox dye. Following the manufacturer’s instructions, all kit components were equilibrated to room temperature and briefly centrifuged all tubes before opening. A 10 µL of the samples, assay buffer as sample blank, and reconstituted positive control solution (catalase) as positive control were loaded into wells of the 96-well plate. Afterward, 50 µM H_2_O_2_ substrate was prepared, and 90 µL of the reagent was added to the samples, sample blank, and positive control wells. The microplate was shaken to mix and incubated at room temperature for 30 min. A four-point standard curve was prepared by serial concentration, mixing 400 µM H_2_O_2_ reagent with dH_2_O. Next, 10 µL of the standard was transferred to separate wells of the plate, and 90 µL assay buffers were added to the standards. At the end of 30 min of incubation, the detection reagent (100 µL) was added and incubated for 10 min at room temperature. Lastly, the optical density was read at 570 nm using a microplate reader (Multiskan ^TM^ GO, Thermo Scientific ^TM^, Waltham, MA, USA), and the *CAT* activity was calculated from the standard curve.

#### 2.4.2. Superoxide Dismutase Activity

EnzyChrom^TM^ Superoxide Dismutase Assay Kit (ESOD-100, BioAssay Systems, Hayward, CA, USA) was used to determine plasma *SOD* activity. The assay provides superoxide through a xanthin oxidase catalyzed reaction, and superoxide reacts with a specific dye (WST-1) to form a colored product. Based on the protocol supplied by the manufacturer, all reagents were prepared, and 20 µL samples were loaded into a 96-well plate. A 3 U/mL *SOD* enzyme standard was made, serially diluted, and 20 µL of each diluent was transferred to separated wells. The working reagent was prepared by mixing assay buffer, xanthine, and WST-1 dye. A 160 µL working reagent was transferred to each well and mixed. In the last step, 20 µL diluted xanthine oxidase enzyme (1:20 in diluent) was added to each assay well and mixed. The assay well plate was read at 450 nm immediately, and, then, after 60 min, dark incubation at room temperature was performed using a microplate reader (Multiskan GO, Thermo Scientific, Waltham, MA, USA). The standard curve was used to calculate the *SOD* concentration in the samples.

#### 2.4.3. Glutathione Peroxidase Activity

Plasma *GPx* activity was determined using EnzyChrom^TM^ Glutathione Peroxidase Assay Kit (EGPx-100, BioAssay Systems, Hayward, CA, USA). The assay is based on the direct measures of NADPH consumption in the enzyme-coupled reactions. According to the manufacturer’s protocol, reagents were prepared/reconstituted, and standards were prepared by mixing 6 mM NADPH with H_2_O. A 10 µL of each diluted standard was loaded to separated wells of a 96-well plate, followed by adding 190 µL assay buffers to each standard well. A 10 µL of each sample was transferred into wells, and background control wells were prepared by adding a 10 µL assay buffer to each well. The working reagent was prepared by mixing assay buffer, glutathione, NADPH, and glutathione reductase enzyme. A 90 µL working reagent was added to the sample and control wells and mixed. In the last step, 100 µL diluted peroxide solution was added to the sample and control wells and mixed. The optical density was read immediately, and, after 4 min at 340 nm, a microplate reader (Multiskan GO, Thermo Scientific, USA) was used. The standard curve was used to calculate the *GPx* activity of samples.

### 2.5. Hepatic Antioxidant Genes Expression

The RNA was extracted from the liver tissue following the manufacturer’s instructions of the NucleoSpin^®^ RNA Plus kit (MACHEREY-NAGEL, Allentown, PA, USA). The gDNA was removed through the lysate filtration using a NucleoSpin^®^ gDNA Removal Column (MACHEREY-NAGEL, Allentown, USA). Based on the instructions of the manufacturer, the RNA was purified using a NucleoSpin^®^ RNA Plus Column (MACHEREY-NAGEL, Allentown, USA). The ultraviolet-visible spectroscopy was used to determine the concentration and purity of RNA by using a spectrophotometer (Multiskan GO, Thermo Scientific, USA). The RNA was then converted into complementary DNA (cDNA) using a cDNA synthesis kit (Biotechrabbit, Hennigsdorf, Germany), following the manufacturer’s instructions.

Real-time PCR was conducted using a LightCycler^®^ 480 qPCR system (Roche Molecular Systems, Indianapolis, IN, USA). The Glyceraldehyde-3-phosphate dehydrogenase (*GAPDH*) was used as a housekeeping gene [19]. A qPCR master mix (20 µL) was prepared using a CAPITALTM qPCR Green Mix, 4× (Biotechrabbit, Hennigsdorf, Germany). The master mix contained 5 µL of SYBR Green Master Mix, 1 µL of each 200 nm forward and reverse primers, 1 µL of template cDNA, and 12 µL of RNase-free water.

The qPCR cycling condition was programmed as follows. The initial denaturation temperature was 95 °C for 2.5 min, followed by 45 cycles of denaturation at 95 °C for 15 sec and annealing for 30 sec at 60 °C. The final extension for melt analysis was based on the instrument instruction, LightCycler^®^ 480 qPCR system (Roche Molecular Systems, Indianapolis, IN, USA). In order to confirm the specificity of the amplification, a melting curve analysis was performed at the end of the amplification cycle. The relative gene expression was quantified following the Livak and Schmittgen [20] recommendation, based on housekeeping gene amplification. The targeted gene primer sequences are presented in Table 3.

### 2.6. Plasma Lipid Profile

Plasma was separated by centrifugation at 3500× *g* at 4 °C for 15 min [21]. The plasma was harvested in the 1.5 mL tube and stored at −80 °C until analysis. The analysis of plasma TC, triglycerides (TG), HDL, and low-density lipoprotein (LDL) were determined on a Dimension^®^ Xpand^®^ Plus Integrated Chemistry System (Siemens Healthcare Diagnostics, Deerfield, IL, USA). In addition, the very-low-density lipoprotein (VLDL) concentration was calculated by dividing plasma TG by 5 [22].

### 2.7. Proximate Analysis of Breast Meat

Proximate analysis was performed for feed and digesta as described in AOAC (AOAC, 1995). Briefly, the dry moisture content was determined by drying the samples at 105 °C for 24 h. The ash contents were determined by the combustion of samples at 550 °C in a muffle furnace for 6 h. The Kjeldahl method determined the CP content using Foss Digestor™ System (Foss Analytical, Denmark) for sample digestion and Foss Kjeltec 2300 (Foss Analytical, Denmark) for distillation and titration. The EE content was determined using the Soxtec system (Tecator, Sweden).

### 2.8. Meat Quality Determination

#### 2.8.1. Meat pH

A pH meter (Mettler Toledo, AG 8603, Switzerland) was used to measure the pH of breast muscle. Approximately 0.5 g of breast meat was weighed, crushed, and homogenized for around 20 sec in 10 mL ice-cold distilled water using the homogenizer (Wiggen Hauser^®^ D-500, Germany), and the pH of each sample was recorded.

#### 2.8.2. Meat Color

The meat samples were removed from the packaging and allowed to bloom in the air for 20 min before color measurement. Meat color measurement was conducted using a Color Flex Spectrophotometer (Hunter Lab Reston, VA, USA), using the International Commission on Illumination (CIE) color values. The instrument was calibrated at a 400–700 nm wavelength to express the meat color data. Three measurements were taken at three separate locations for each sample. The color for each sample was expressed according to CIE values for L* (lightness), a* (redness), and b* (yellowness).

#### 2.8.3. Water Holding Capacity of Meat

The water holding capacity (WHC) was determined in terms of drip and cooking losses, as described by Honikel [23]. 

Around 30 g of fresh meat sample was weighed (initial weight) and vacuum-packed to determine the drip loss. The sample was kept in the chiller at 4 °C for 24 h and seven days. The samples were then patted dry using a paper towel and weighed again (final weight). The calculation was performed as follows:(2)$$\mathrm{Drip}\;\mathrm{loss}\%=\frac{(\mathrm{initial}\;\mathrm{weight}\;-\;\mathrm{final}\;\mathrm{weight})}{(\mathrm{initial}\;\mathrm{weight})}\times 100$$

The breast meat sample was kept in a chiller at 4 °C for 24 h and seven days for the cooking loss determination. The sample was weighed and recorded (W1) and transferred into a preheated water bath set at 80 °C for 20 min. After cooling, the sample was patted dry using a paper towel without squeezing, reweighed, and recorded (W2). The cooking loss was calculated based on the following equation:(3)$$\mathrm{Cooking}\;\mathrm{loss}\%\;=\frac{\left(W1-\;W2\right)\;}{W1}\times 100$$
where, W1= initial sample weight before cooking, W2= sample weight after cooking.

#### 2.8.4. Meat Tenderness or Shear Force

The tenderness or shear force examination is based on the mechanical force (kg) required to shear the cooked meat’s muscle fibers. The previously cooked samples for cooking loss determination (at 80 °C for 20 min) were used to measure meat tenderness. The samples were cut as; 1 cm width, 1 cm thickness, and 2 cm length, matching the muscle fibers’ direction [24]. The method was applied according to the procedures described by [25]. The sample was sheared perpendicularly to the muscle fibers using TA.HD plus^®^ texture analyzer (Stable Micro System, Surrey, UK). The analyzer was equipped with a Volodkevitch blade set. The apparatus was standardized at 15 mm return distance for height, 5 kg for weight, and the blade speed was set at 10 mm/sec. The shear force values are noted as the average of all samples value for each sample.

### 2.9. Statistical Analysis

The general linear model (GLM), by one-way ANOVA of the statistical analysis system (SAS), was used for the statistical analysis. Duncan’s multiple range test was used to compare the significant difference between the treatment groups at *p* < 0.05. The orthogonal polynomial contrast of SAS was used to determine the linear and quadratic effects of dietary increasing brown and green seaweed inclusion levels. The statistical model for the experiment was Yijk = µ + Tij + Eijk. Whereas, Yijk = dependent variable, µ = general mean, Tij = effect of dietary treatment, and Eijk = experimental error.

## 3. Results

### 3.1. Antioxidant Property of Brown and Green Seaweed

The result of the DPPH scavenging potential of seaweed is presented in Figure 1. The findings showed that both BS and GS exhibited remarkable antioxidant activity. Whereas, the maximum antioxidant activity was recorded by BS (55.19%), which was also significantly higher (*p* < 0.05) than the GS (25.74%).

### 3.2. Plasma Antioxidant Enzymes Activities

The effects of seaweed on broiler chickens’ blood plasma antioxidant enzyme activities are presented in Table 4. The results showed that various brown and green seaweed supplementations had no significant effects on blood plasma *CAT*, *SOD*, or *GPx* enzyme activities. 

### 3.3. Hepatic Antioxidant Genes Expression

The mRNA expression of the hepatic *SOD1*, *GPX1*, and *CAT* genes of broiler chickens fed various levels of brown and green seaweed are presented in Table 5. The result showed that the hepatic *SOD1* mRNA expression was higher (*p* < 0.05) for birds fed 0.50% and 0.75% BS than in the NC and PC groups. On the other hand, no difference (*p* > 0.05) was observed in the *SOD1* mRNA expression for different GS groups compared to the control groups. Meanwhile, brown and green seaweed had no effects (*p* > 0.05) on the hepatic *GPX1* and *CAT* genes’ mRNA expression. 

### 3.4. Plasma Lipid Profile

The effects of BS and GS on blood plasma TC, TG, HDL, LDL, and VLD of broiler chicken are presented in Table 6. Various brown and green seaweed levels had no effects (*p* > 0.05) on plasma TG, LDL, and VLDL levels. However, the TC and HDL levels were significantly higher (*p* < 0.05) for birds fed 0.75 and 1% BS compared to the NC groups. No significant difference was observed in plasma TC and HDL for birds fed GS supplemented feed compared to the NC and PC groups.

### 3.5. Proximate Analysis of Breast Meat

The chemical composition of breast meat in broiler chickens fed seaweed is presented in Table 7. Birds fed 0.25 and 0.50% BS and 0.25, 0.75 and 1% GS had higher (*p* < 0.05) breast meat CP content compared to the NC birds. In contrast, the EE content of breast meat was significantly lower for birds fed 0.50% BS and 0.75 and 1.25% GS. No differences (*p* > 0.05) were observed in the moisture and ash contents of the breast meat among the dietary treatments.

### 3.6. Meat Quality

The effects of BS and GS on breast meat color, pH, drip loss, cooking loss, and shear force after days one and seven of storage are presented in Table 8.

The BS and GS had no significant effects on breast meat pH value, shear force, cooking loss, and drip loss percentage after days one and seven of storage. The cooking loss for the 1.25% BS and 0.25% GS groups and drip loss for the 0.75% BS and 1.25% GS were significantly decreased linearly and quadratically compared to the NC group after day seven. Chickens fed 0.25%, 0.50%, 1%, and 1.25% BS and GS had significantly lower meat lightness than the NC group. In addition, the day one meat color yellowness was lower (*p* < 0.05) for chickens fed 0.75% BS than the NC group. The color yellowness was higher (*p* < 0.05) for chickens fed 1% GS than the NC group. There was no significant difference in breast meat color parameters after seven days among the dietary treatments.

## 4. Discussion

### 4.1. Antioxidant Properties of Brown and Green Seaweed

DPPH is a simple, rapid, and inexpensive antioxidant capacity measurement technique. This assay evaluates the overall antioxidant capacity by determining the ability of compounds to act as free radical scavengers or H atoms donors [26]. This study showed that both BS and GS show prominent antioxidant activity. A series of earlier research findings have reported that seaweed is a source of natural antioxidants. The presence of numerous biologically active compounds, such as sulfated polysaccharides, carotenoids, β-carotene, vitamin C, and α- tocopherol, could contribute to the antioxidant efficiency of seaweed [13,14,27].

The result of the current study also showed that the BS recorded the maximum antioxidant activity. Fucoidan is the main polysaccharide present in BS that can reach up to 20 to 200 g kg^−1^ of BS dry weight [28]. Research has shown that fucoidan polysaccharides showed considerable antioxidant activity [27]. Furthermore, the fucoxanthin carotenoid is another voluble biological compound of BS that can reach up to 5000 mg/kg of algal mass [1,29]. It is reported that fucoxanthin has exhibited excellent antioxidant properties that can protect cells against oxidation-induced damage [12,15]. Moreover, vitamin E is one of the most important fat-soluble vitamins of seaweed with a strong antioxidant capability [12,30,31], whereas it had shown that the vitamin E content of BS is higher than the GS [31]. Additionally, the phenolic compounds of seaweed are also effective primary antioxidants due to their capability to donate their H atoms to the free radicals to make them stable [12]. 

### 4.2. Plasma Antioxidant Enzymes Activities

Numerous free radicals, including hydrogen peroxide, superoxide anion, singlet oxygen, hydroxyl radical, peroxyl radicals, alkoxy radicals, and reactive nitrogen species, can damage mitochondria, DNA, and almost all organic molecules, such as carbohydrates, lipids, and proteins [8].

Antioxidants consist of enzymatic and non-enzymatic antioxidants. The main enzymatic antioxidants are *SOD*, *CAT*, and *GPx* [26]. *CAT*, *SOD*, and *GPx* are the main antioxidant enzymes in chicken plasma that transform free radicals into non-radical and non-toxic products. *CAT* enzyme catalysis is the decomposition of hydrogen peroxide (H_2_O_2_) to water and O_2_. *SOD* enzyme catalyzes the dismutation of superoxide into O_2_ and H_2_O_2_. In contrast, the *GP_X_* prevents the peroxidation of cellular membranes lipid by removing free peroxide in the cell [32,33]. Natural antioxidants have been of great in interest in recent years. Seaweed contains dietary antioxidants that can be used as medicinal and preventative agents [6,10,11]. Numerous in vitro studies have revealed that seaweed has reactive oxygen species (ROS) scavenging properties [34]. The current study showed that different levels of brown and green seaweed had no significant effects on birds’ plasma *CAT*, *SOD*, and *GPx* enzyme activities. Consistent with the present study, Ramirez-Higuera et al. [35] showed that administration of BS *Lessonia trabeculata* and GS *Ulva linza* at 400 mg/kg^−1^ body weight in Wistar rats did not significantly affect the *CAT*, *SOD*, and *GPx* enzymes activities in liver tissues. Similarly, Maheswari et al. [36] reported a 2.5% inclusion of a red and brown seaweed blend (*Kappaphycus alvarezii*, *Gracilaria salicornia* and *Turbinaria conoides*) (1:1:1) in lactating Murrah buffaloes’ diet did not affect their plasma *SOD* enzyme activity. Whereas, the results obtained by Gumus et al. [16] have stated that 100 and 200 mg/kg inclusion levels of BS fucoxanthin extract in broiler chickens increased the *CAT* and *SOD* activities in the liver, breast, and drumstick tissues. This result indicates that the antioxidant effects of fucoxanthin extract may be higher compared to the whole seaweed.

### 4.3. Hepatic Antioxidant Genes Expression

Previous research has indicated that seaweed bioactive compounds can act as antioxidants. Thereby, seaweed may protect cells and tissues from the adverse effects of free radicals and singlet oxygen [37,38]. The current study showed that the hepatic *SOD1* mRNA expression was higher for birds fed 0.50% and 0.75% BS than in the NC and PC groups. This finding can be linked to the result from this study—that, due to the DPPH scavenging potential, BS recorded significantly higher antioxidant activity than GS. The findings reported by Gumus et al. [16] showed that 100 and 200 mg/kg of fucoxanthin extract in broiler diets increased the *SOD1* level in the liver, breast, and drumstick tissues. Fucoxanthin carotenoid is a voluble biological compound of BS that can reach up to 5000 mg/kg of algal mass [1,29]. It was reported that fucoxanthin exhibited excellent antioxidant properties that protect cells against oxidation-induced damage [12,15].

### 4.4. Plasma Lipid Profile

The result showed that different brown and green seaweed levels did not affect plasma TG, LDL, and VLD levels. Meanwhile, no significant effects were observed on plasma TC and HDL for birds fed all levels of GS and 0.25, 0.50, and 1.25% BS. Nevertheless, 0.75 and 1% BS had higher TC and HDL levels. These findings are consistent with those of Choi et al. [39], who reported that the inclusion of 0.5% BS had no effects on plasma TG, TC, and HDL levels in broiler chickens. The results contradict the findings from Kumar [40] that include 1, 2, 3, and 4% BS *Sargassum* in the broiler diet-decreased plasma TC level, while increasing the TG level, compared to the control birds. These results for plasma lipid profile are also inconsistent with earlier studies on rats, which found that various brown, green, and red seaweed levels decreased animals’ plasma TC, TG, and LDL levels [27,41,42]. The differences in the animal systems, seaweed species, and basal feeds can explain these contradictory results.

### 4.5. Proximate Analysis of Breast Meat

The results demonstrated that 0.25 and 0.50% BS and 0.25, 0.75, and 1% GS had improved the CP content of broiler breast meat. This improvement in CP may be due to the high protein and amino acid contents of seaweed. The amino acid value of seaweed proteins is higher than vegetables and cereals [27]. Moreover, our results showed that 0.50% BS and 0.75 and 1.25% GS reduced breast meat’s EE content. The biological compounds present in seaweed may affect lipid metabolism [43]. It has been determined that brown, green, and red seaweed showed cholesterol-lowering activities [6]. Polysaccharides of seaweed absorb substances, including cholesterol, and they remove them from the digestive tract [27].

### 4.6. Meat Quality 

For the meat industry, meat quality is a term used to describe overall meat characteristics [44]. Apart from the health consequences, the meat quality obtained from broiler chicken is also an essential parameter. Lipid oxidation can negatively affect broiler meat’s color, flavor, texture, and nutritional value [9]. Besides adversely affecting meat quality, lipid peroxidation also constitutes a health risk in the event of the consumption of meat that has undergone lipid peroxidation [16]. Thus, incorporating natural dietary antioxidants can be more effective and practical in controlling lipid-oxidation-related products and providing nutritious and healthy products to consumers [9,16]. The current study results showed that different levels of BS and GS had no significant effects on various meat quality indicators, such as pH value, cooking loss, and drip loss after days one and seven of storage compared to the NC group. Whereas, the chickens fed different BS and GS feed levels had lower lightness than the NC group birds meat on day one. There were no significant differences in breast meat color lightness, redness, and yellowness after seven days of storage among dietary treatments. 

Little published research is available to show the effect of seaweed on meat quality in broiler chickens. However, an earlier report showed that 100 g/kg fucoxanthin supplements from BS had no effects on broiler meat color’s lightness and redness parameters on the first day of storage, while the yellowness was significantly increased. Simultaneously, the fucoxanthin did not affect the pH value of birds’ meat [16]. In another study, Abudabos et al. [45] reported that the inclusion of 1 and 3% of GS in broiler chicken feeding had no effect on broiler breast meat color parameters. 

## 5. Conclusions

BS and GS exhibited remarkable antioxidant activity. Brown and green seaweed up to 1.25% inclusion levels had no significant effects on blood plasma antioxidant enzyme activities. In contrast, the hepatic *SOD1* gene mRNA expression was increased for birds fed 0.50% and 0.75% of BS. At the same time, BS and GS had no effects on blood plasma TG, LDL, and VLDL levels in broiler chickens. The EE content of breast meat was decreased for birds fed 0.50% BS and 0.75 and 1.25% GS-containing feed. Meanwhile, birds fed 0.25 and 0.50% BS, as well as 0.25, 0.75, and 1% GS, had increased CP content. In contrast, no differences were observed among the dietary treatments for the breast meat pH, shear force, cooking loss, and drip loss percentage after days one and seven of storage. The current research findings are useful for further studies investigating the potential antioxidant activity within the blood and several tissues. Discoveries from this research are also helpful for further studies to focus on the effects of seaweed on broiler meat amino acids and lipid profiles.

## Figures and Tables

**Figure 1 animals-13-01582-f001:**
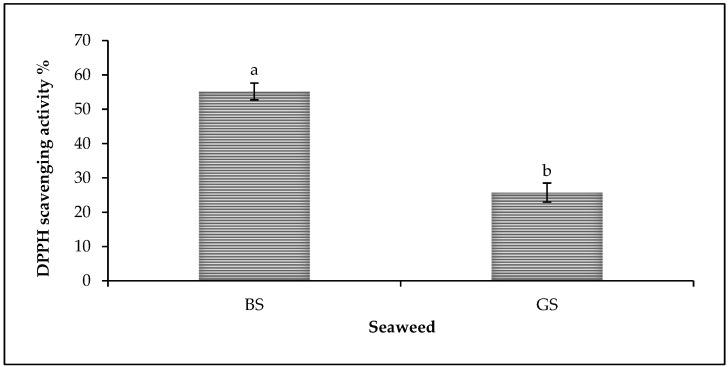
DPPH radical scavenging activities percentage of seaweed. BS = brown seaweed, GS = green seaweed. ^a,b^ Different letters on standard error bars indicate a significant difference between seaweed types (*p* < 0.05). Data are shown as means and standard error (*n* = 6).

**Table 1 animals-13-01582-t001:** Ingredient composition of the starter period (day 1–22) diet.

Ingredients (%)	Dietary Treatments ^1^											
	NC	PC	BS 0.25	BS 0.50	BS 0.75	BS 1	BS 1.25	GS 0.25	GS 0.50	GS 0.75	GS 1	GS 1.25
Corn	46.0	46.0	46.0	46.0	46.0	46.0	46.0	46.0	46.0	46.0	46.0	46.0
Soybean meal	40.0	40.0	39.8	39.5	39.3	39.0	38.8	39.8	39.5	39.3	39.0	38.8
Wheat pollard	5.0	5.0	5.0	5.0	5.0	5.0	5.0	5.0	5.0	5.0	5.0	5.0
Palm oil	4.0	4.0	4.0	4.0	4.0	4.0	4.0	4.0	4.0	4.0	4.0	4.0
L-Lysine ^2^	0.20	0.20	0.20	0.20	0.20	0.20	0.20	0.20	0.20	0.20	0.20	0.20
DL-Methionine ^3^	0.40	0.40	0.40	0.40	0.40	0.40	0.40	0.40	0.40	0.40	0.40	0.40
DCP ^4^	2.60	2.60	2.60	2.60	2.60	2.60	2.60	2.60	2.60	2.60	2.60	2.60
Calcium carbonate	0.80	0.80	0.80	0.80	0.80	0.80	0.80	0.80	0.80	0.80	0.80	0.80
Choline chloride	0.20	0.20	0.20	0.20	0.20	0.20	0.20	0.20	0.20	0.20	0.20	0.20
Salt	0.30	0.30	0.30	0.30	0.30	0.30	0.30	0.30	0.30	0.30	0.30	0.30
Mineral mix ^5^	0.15	0.15	0.15	0.15	0.15	0.15	0.15	0.15	0.15	0.15	0.15	0.15
Vitamin mix ^6^	0.15	0.15	0.15	0.15	0.15	0.15	0.15	0.15	0.15	0.15	0.15	0.15
Antioxidants	0.10	0.10	0.10	0.10	0.10	0.10	0.10	0.10	0.10	0.10	0.10	0.10
Toxin binder	0.10	0.10	0.10	0.10	0.10	0.10	0.10	0.10	0.10	0.10	0.10	0.10
Seaweed	-	-	0.25	0.50	0.75	1.00	1.25	0.25	0.50	0.75	1.00	1.25
Vitamin E	-	0.01	-	-	-	-	-	-	-	-	-	-
Total	100	100	100	100	100	100	100	100	100	100	100	100
Calculated analysis ^7^												
ME (kcal/kg) ^8^	3040.2	3039.9	3041.0	3041.9	3042.7	3043.6	3044.5	3040.7	3041.3	3041.9	3042.5	3043.0
Protein	21.95	21.95	21.94	21.91	21.90	21.89	21.87	21.93	21.90	21.87	21.85	21.82
Fat	5.98	5.98	5.98	5.98	5.98	5.98	5.98	5.98	5.98	5.98	5.97	5.97
Fiber	4.34	4.34	4.33	4.31	4.31	4.29	4.28	4.32	4.31	4.30	4.29	4.28
Calcium	0.83	0.83	0.83	0.83	0.83	0.83	0.83	0.83	0.83	0.83	0.83	0.83
Total phosphorous	1.01	1.01	1.01	1.01	1.01	1.00	1.00	1.01	1.01	1.00	1.00	1.00
Available phosphorus	0.50	0.50	0.50	0.50	0.50	0.50	0.50	0.50	0.50	0.50	0.50	0.50

^1^ Dietary treatments: NC (negative control) = basal diet, PC (positive control) = basal diet + vitamin E (100 mg/kg feed), BS 0.25 = basal diet + 0.25% brown seaweed, BS 0.50 = basal diet + 0.50% brown seaweed, BS 0.75 = basal diet + 0.75% brown seaweed, BS 1 = basal diet + 1% brown seaweed, BS 1.25 = basal diet + 1.25% brown seaweed, GS 0.25 = basal diet + 0.25% green seaweed, GS 0.50 = basal diet + 0.50% green seaweed, GS 0.75 = basal diet + 0.75% green seaweed, GS 1 = basal diet + 1% green seaweed, GS 1.25 = basal diet + 1.25% green seaweed. ^2^ L-Lysine 78.8% (minimum). ^3^ DL-Methionine 99%. ^4^ Dicalcium phosphate. ^5^ Mineral mix [provided per Kg of the product (mineral mix)]: Selenium 0.20 g; iron 80.0 g; manganese 100.0 g; zinc 80.0 g; copper 15.0 g; potassium 4.0 g; sodium 1.50 g; iodine 1.0 g and cobalt 0.25 g. ^6^ Vitamin premix [provided per Kg of the product (vitamin premix)]: Vitamin A 35.0 MIU; vitamin D3 9.0 MIU; vitamin E 90.0 g; vitamin K3 6.0 g; vitamin B1 7.0 g; vitamin b2 22.0 g; vitamin B6 12.0 g; vitamin B12 0.070 g; pantothenic acid 35.0 g; nicotinic acid 120.0 g; folic acid 3.0 g; biotin 300.000 mg; phytase 25,000.0 FTU cobalamin 0.05 mg; thiamine 1.43 mg; riboflavin 3.44 mg; folic acid 0.56 mg; biotin 0.05 mg; pantothenic acid 6.46 mg; niacin 40.17 mg and pyridoxine 2.29 mg. ^7^ The diets were formulated using FeedLIVE software. ^8^ Metabolizable energy.

**Table 2 animals-13-01582-t002:** Ingredient composition of the finisher period (day 22–42) diet.

Ingredients (%)	Dietary Treatments ^1^											
	NC	PC	BS 0.25	BS 0.50	BS 0.75	BS 1	BS 1.25	GS 0.25	GS 0.50	GS 0.75	GS 1	GS 1.25
Corn	52.0	52.0	52.0	52.0	52.0	52.0	52.0	52.0	52.0	52.0	52.0	52.0
Soybean meal	32.0	32.0	31.8	31.5	31.3	31.0	30.8	31.8	31.5	31.3	31.0	30.8
Wheat pollard	6.0	6.0	6.0	6.0	6.0	6.0	6.0	6.0	6.0	6.0	6.0	6.0
Palm oil	5.10	5.10	5.10	5.10	5.10	5.10	5.10	5.10	5.10	5.10	5.10	5.10
L-Lysine ^2^	0.20	0.20	0.20	0.20	0.20	0.20	0.20	0.20	0.20	0.20	0.20	0.20
DL-Methionine ^3^	0.30	0.30	0.30	0.30	0.30	0.30	0.30	0.30	0.30	0.30	0.30	0.30
DCP ^4^	2.40	2.40	2.40	2.40	2.40	2.40	2.40	2.40	2.40	2.40	2.40	2.40
Calcium carbonate	1.0	1.0	1.0	1.0	1.0	1.0	1.0	1.0	1.0	1.0	1.0	1.0
Choline chloride	0.20	0.20	0.20	0.20	0.20	0.20	0.20	0.20	0.20	0.20	0.20	0.20
Salt	0.30	0.30	0.30	0.30	0.30	0.30	0.30	0.30	0.30	0.30	0.30	0.30
Mineral mix ^5^	0.15	0.15	0.15	0.15	0.15	0.15	0.15	0.15	0.15	0.15	0.15	0.15
Vitamin mix ^6^	0.15	0.15	0.15	0.15	0.15	0.15	0.15	0.15	0.15	0.15	0.15	0.15
Antioxidants	0.10	0.10	0.10	0.10	0.10	0.10	0.10	0.10	0.10	0.10	0.10	0.10
Toxin binder	0.10	0.10	0.10	0.10	0.10	0.10	0.10	0.10	0.10	0.10	0.10	0.10
Seaweed	-	-	0.25	0.50	0.75	1.00	1.25	0.25	0.50	0.75	1.00	1.25
Vitamin E	-	0.01	-	-	-	-	-	-	-	-	-	-
Total	100	100	100	100	100	100	100	100	100	100	100	100
Calculated analysis ^7^												
ME (kcal/kg) ^8^	3149.8	3149.5	3150.7	3151.5	3152.4	3153.3	3154.1	3150.4	3150.9	3151.6	3152.1	3152.7
Protein	19.06	19.06	19.05	19.03	19.01	19.00	18.98	19.04	19.01	18.98	18.96	18.93
Fat	7.19	7.19	7.19	7.19	7.19	7.19	7.19	7.19	7.19	7.18	7.18	7.18
Fiber	4.00	4.00	3.99	3.98	3.97	3.96	3.95	3.99	3.98	3.97	3.96	3.94
Calcium	0.85	0.85	0.85	0.85	0.85	0.85	0.85	0.85	0.85	0.85	0.85	0.85
Total phosphorous	0.94	0.94	0.94	0.94	0.94	0.94	0.94	0.94	0.94	0.94	0.94	0.94
Available phosphorus	0.47	0.47	0.47	0.47	0.47	0.47	0.47	0.47	0.47	0.47	0.47	0.47

^1^ Dietary treatments: NC (negative control) = basal diet, PC (positive control) = basal diet + vitamin E (100 mg/kg feed), BS 0.25 = basal diet + 0.25% brown seaweed, BS 0.50 = basal diet + 0.50% brown seaweed, BS 0.75 = basal diet + 0.75% brown seaweed, BS 1 = basal diet + 1% brown seaweed, BS 1.25 = basal diet + 1.25% brown seaweed, GS 0.25 = basal diet + 0.25% green seaweed, GS 0.50 = basal diet + 0.50% green seaweed, GS 0.75 = basal diet + 0.75% green seaweed, GS 1 = basal diet + 1% green seaweed, GS 1.25 = basal diet + 1.25% green seaweed. ^2^ L-Lysine (minimum) 78.8%. ^3^ DL-Methionine 99%. ^4^ Dicalcium phosphate. ^5^ Mineral mix [provided per Kg of the product (mineral mix)]: Selenium 0.20 g; iron 80.0 g; manganese 100.0 g; zinc 80.0 g; copper 15.0 g; potassium 4.0 g; sodium 1.50 g; iodine 1.0 g and cobalt 0.25 g. ^6^ Vitamin premix [provided per Kg of the product (Vitamin premix)]: Vitamin A 35.0 MIU; vitamin D3 9.0 MIU; vitamin E 90.0 g; vitamin K3 6.0 g; vitamin B1 7.0 g; vitamin b2 22.0 g; vitamin B6 12.0 g; vitamin B12 0.070 g; pantothenic acid 35.0 g; nicotinic acid 120.0 g; folic acid 3.0 g; biotin 300.000 mg; phytase 25,000.0 FTU cobalamin 0.05 mg; thiamine 1.43 mg; riboflavin 3.44 mg; folic acid 0.56 mg; biotin 0.05 mg; pantothenic acid 6.46 mg; n7iacin 40.17 mg and pyridoxine 2.29 mg. ^7^ The diets were formulated using FeedLIVE software. ^8^ Metabolizable energy.

**Table 3 animals-13-01582-t003:** The primer sequences of the housekeeping and target genes.

Target Gene	Primer Sequence 5′-3′		bp	Accession No.
*SOD1*	F-CACTGCATCATTGGCCGTACCA	R-GCTTGCACACGGAAGAGCAAGT	224	NM_205064.1
*GPX1*	F-GCTGTTCGCCTTCCTGAGAG	R-GTTCCAGGAGACGTCGTTGC	118	NM_001277853.2
*CAT*	F-TGGCGGTAGGAGTCTGGTCT	R-GTCCCGTCCGTCAGCCATTT	139	NM_001031215.2
*GAPDH*	F-CTGGCAAAGTCCAAGTGGTG	R-AGCACCACCCTTCAGATGAG	275	NM_204305.1

F = Forward, R: Reverse. bp (base pair) = Product size. *SOD1* = Superoxide dismutase 1, *GPX1* = Glutathione peroxidase 1, *CAT* = Catalase, *GAPDH* = Glyceraldehyde-3-phosphate dehydrogenase.

**Table 4 animals-13-01582-t004:** Effects of different brown and green seaweed levels on plasma antioxidant enzyme activities in broiler chickens.

Enzymes ^1^	Dietary Treatments ^2^												SEM ^3^	*p*-Values	Contrast *p*-Values ^4^	
	NC	PC	BS 0.25	BS 0.50	BS 0.75	BS 1	BS 1.25	GS 0.25	GS 0.50	GS 0.75	GS 1	GS 1.25			Line.	Quad.
*CAT* (U/L)	4.11	3.71	3.73	3.73	3.65	3.84	4.16	3.46	3.97	3.96	3.92	4.00	0.21	0.6222	0.3072	0.1506
*SOD* (U/mL)	3.10	3.11	3.14	3.08	3.11	3.09	3.15	3.04	3.10	3.12	3.14	3.11	0.04	0.9395	0.4899	0.8876
*GPx* (U/L)	0.48	0.48	0.51	0.46	0.49	0.47	0.43	0.51	0.44	0.46	0.46	0.47	0.03	0.6493	0.5120	0.6833

^1^ Enzymes: *CAT* = catalase. One unit is the amount of *CAT* that decomposes 1 µmole of H_2_O_2_ per min at pH 7.0 and room temperature; *SOD* = superoxide dismutase. One unit is the amount of *SOD* that catalyze 1 µmole of superoxide into O_2_ and H_2_O_2_ per min under the condition of the assay; *GPx* = glutathione peroxidase. One unit is the amount of *GP_X_* that produces 1 µmole of glutathione disulfide per min at pH 7.6 and room temperature. ^2^ Dietary treatments: NC (negative control) = basal diet, PC (positive control) = basal diet + vitamin E (100 mg/kg feed), BS 0.25 = basal diet + 0.25% brown seaweed, BS 0.50 = basal diet + 0.50% brown seaweed, BS 0.75 = basal diet + 0.75% brown seaweed, BS 1 = basal diet + 1% brown seaweed, BS 1.25 = basal diet + 1.25% brown seaweed, GS 0.25 = basal diet + 0.25% green seaweed, GS 0.50 = basal diet + 0.50% green seaweed, GS 0.75 = basal diet + 0.75% green seaweed, GS 1 = basal diet + 1% green seaweed, GS 1.25 = basal diet + 1.25% green seaweed. ^3^ SEM = standard error of means. ^4^ Contrast *p*-Values = orthogonal polynomial contrasts of dietary increasing brown and green seaweed inclusion levels (0.0 to 1.25%).

**Table 5 animals-13-01582-t005:** Effects of brown and green seaweed on *SOD*, *GPX1*, and *CAT* mRNA expression in broiler chickens.

Parameters ^1^ (mRNA Fold Change)	Dietary Treatments ^2^												SEM ^3^	*p*-Values	Contrast *p*-Values ^4^	
	NC	PC	BS 0.25	BS 0.50	BS 0.75	BS 1	BS 1.25	GS 0.25	GS 0.50	GS 0.75	GS 1	GS 1.25			Line.	Quad.
*SOD1*	1 ^c^	1.053 ^c^	1.13 ^b,c^	1.517 ^a,b^	1.604 ^a^	0.798 ^c^	1.174 ^a,b,c^	0.935 ^c^	1.208 ^a,b,c^	1.176 ^a,b,c^	0.992 ^c^	1.211 ^a,b,c^	0.049	0.0193	0.5515	0.0597
*GPX1*	1	1.038	1.096	0.763	1.020	0.979	0.972	0.834	0.643	1.013	0.779	0.912	0.046	0.4557	0.4080	0.3860
*CAT*	1	1.002	1.301	1.050	1.007	1.034	0.793	1.029	0.847	1.074	1.064	0.858	0.041	0.7925	0.2322	0.5524

^a,b,c^ Means with different superscripts in the same row indicates significant difference (*p* < 0.05). ^1^ Parameters: *SOD1* = superoxide dismutase 1, *GPX1* = glutathione peroxidase 1, *CAT* = catalase. ^2^ Dietary treatments: NC (negative control) = basal diet, PC (positive control) = basal diet + vitamin E (100 mg/kg feed), BS 0.25 = basal diet + 0.25% brown seaweed, BS 0.50 = basal diet + 0.50% brown seaweed, BS 0.75 = basal diet + 0.75% brown seaweed, BS 1 = basal diet + 1% brown seaweed, BS 1.25 = basal diet + 1.25% brown seaweed, GS 0.25 = basal diet + 0.25% green seaweed, GS 0.50 = basal diet + 0.50% green seaweed, GS 0.75 = basal diet + 0.75% green seaweed, GS 1 = basal diet + 1% green seaweed, GS 1.25 = basal diet + 1.25% green seaweed. ^3^ SEM = standard error of means. ^4^ Contrast *p*-values = orthogonal polynomial contrasts of dietary increasing brown and green seaweed inclusion levels (0.0 to 1.25%).

**Table 6 animals-13-01582-t006:** Effects of brown and green seaweed on broiler plasma lipid profile.

Parameters ^1^ (mmol/L)	Dietary Treatments ^2^												SEM ^3^	*p*-Values	Contrast *p*-Values ^4^	
	NC	PC	BS 0.25	BS 0.50	BS 0.75	BS 1	BS 1.25	GS 0.25	GS 0.50	GS 0.75	GS 1	GS 1.25			Line.	Quad.
TC	2.43 ^c^	2.76 ^b,c^	2.9 ^b,c^	2.95 ^b,c^	4.0 ^a^	3.5 ^a,b^	2.98 ^b,c^	3.05 ^b,c^	3.28 ^a,b,c^	3.18 ^a,b,c^	2.9 ^b,c^	2.6 ^b,c^	0.26	0.0287	0.0182	0.1473
TG	0.31	0.29	0.3	0.34	0.37	0.39	0.42	0.54	0.41	0.29	0.45	0.48	0.07	0.3979	0.0555	0.6822
HDL	1.85 ^c^	2.68 ^b,c^	2.35 ^b,c^	2.45 ^b,c^	3.68 ^a^	2.74 ^b^	2.44 ^b,c^	2.5 ^b,c^	2.64 ^b,c^	2.68 ^b,c^	2.42 ^b,c^	2.19 ^b,c^	0.22	0.0067	0.0110	0.0124
LDL	0.57	0.65	0.57	0.58	0.82	0.57	0.58	0.58	0.75	0.6	0.51	0.53	0.09	0.5385	0.1541	0.5914
VLDL	0.06	0.07	0.08	0.08	0.11	0.08	0.06	0.09	0.1	0.06	0.07	0.07	0.01	0.3979	0.0566	0.6360

^a,b,c^ Means with different superscripts in the same row indicate a significant difference (*p* < 0.05). ^1^ Parameters: TC = total cholesterol, TG = triglyceride, HDL = high-density lipoprotein, LDL = low-density lipoprotein, VLDL = very low-density lipoprotein. ^2^ Dietary treatments: NC (negative control) = basal diet, PC (positive control) = basal diet + vitamin E (100 mg/kg feed), BS 0.25 = basal diet + 0.25% brown seaweed, BS 0.50 = basal diet + 0.50% brown seaweed, BS 0.75 = basal diet + 0.75% brown seaweed, BS 1 = basal diet + 1% brown seaweed, BS 1.25 = basal diet + 1.25% brown seaweed, GS 0.25 = basal diet + 0.25% green seaweed, GS 0.50 = basal diet + 0.50% green seaweed, GS 0.75 = basal diet + 0.75% green seaweed, GS 1 = basal diet + 1% green seaweed, GS 1.25 = basal diet + 1.25% green seaweed. ^3^ SEM = standard error of means. ^4^ Contrast *p*-Values = orthogonal polynomial contrasts of dietary increasing brown and green seaweed inclusion levels (0.0 to 1.25%).

**Table 7 animals-13-01582-t007:** Effects of brown and green seaweed on broiler breast meat proximate composition.

Nutrients ^1^ (%)	Dietary Treatments ^2^														Contrast *p*-Values ^4^	
	NC	PC	BS 0.25	BS 0.50	BS 0.75	BS 1	BS 1.25	GS 0.25	GS 0.50	GS 0.75	GS 1	GS 1.25	SEM ^3^	*p*-Values	Line.	Quad.
Moisture	74.43	73.03	74.25	74.03	74.18	74.38	74.17	73.5	73.56	74.71	74.04	74.1	0.40	0.3168	0.0755	0.6850
CP	23.04 ^c^	23.85 ^b,c^	24.51 ^a,b^	24.69 ^a,b^	23.93 ^a,b,c^	23.79 ^b,c^	24.29 ^a,b,c^	24.33 ^a,b^	25.14 ^a^	24.04 ^a,b,c^	24.75 ^a,b^	23.71 ^b,c^	0.38	0.0404	0.0218	0.4461
EE	2.23 ^a^	2.19 ^a,b^	1.99 ^a,b,c^	1.88 ^b,c^	1.93 ^a,b,c^	1.92 ^a,b,c^	2.11 ^a,b^	2.05 ^a,b^	1.96 ^a,b,c^	1.70 ^c^	2.05 ^a,b^	1.87 ^b,c^	0.09	0.0258	0.7959	0.0619
Ash	1.79	1.89	1.68	1.66	1.75	1.67	1.76	1.78	1.74	1.67	1.7	1.64	0.07	0.4587	0.5104	0.2256

^a,b,c^ Means with different superscripts in the same row indicates significant difference (*p* < 0.05). ^1^ Nutrients: CP = crude protein, EE (ether extract). ^2^ Dietary treatments: NC (negative control) = basal diet, PC (positive control) = basal diet + vitamin E (100 mg/kg feed), BS 0.25 = basal diet + 0.25% brown seaweed, BS 0.50 = basal diet + 0.50% brown seaweed, BS 0.75 = basal diet + 0.75% brown seaweed, BS 1 = basal diet + 1% brown seaweed, BS 1.25 = basal diet + 1.25% brown seaweed, GS 0.25 = basal diet + 0.25% green seaweed, GS 0.50 = basal diet + 0.50% green seaweed, GS 0.75 = basal diet + 0.75% green seaweed, GS 1 = basal diet + 1% green seaweed, GS 1.25 = basal diet + 1.25% green seaweed. ^3^ SEM = standard error of means. ^4^ Contrast *p*-Values = orthogonal polynomial contrasts of dietary increasing brown and green seaweed inclusion levels (0.0 to 1.25%).

**Table 8 animals-13-01582-t008:** Effects of brown and green seaweed on broiler breast meat quality.

Parameters	Dietary Treatments ^1^												SEM ^2^	*p*-Values	Contrast *p*-Values ^3^	
	NC	PC	BS 0.25	BS 0.50	BS 0.75	BS 1	BS 1.25	GS 0.25	GS 0.50	GS 0.75	GS 1	GS 1.25			Line.	Quad.
Day 1																
pH	6.02	5.90	5.78	5.87	6.03	5.87	6.05	5.94	5.79	5.97	5.83	5.83	0.07	0.2827	0.7491	0.9708
Drip loss%	1.80	1.55	1.72	1.70	1.44	1.72	1.38	1.46	1.39	1.07	1.23	1.56	0.23	0.6304	0.9737	0.9700
Cooking loss%	28.8	27.21	27.98	25.88	23.62	25.56	23.55	21.02	26.2	22.28	23.04	25.94	28.8	0.0939	0.2273	0.1439
Shear force (g)	978.3	811.5	1112.9	832.9	913.2	971.2	788.7	869.3	1058.6	1016.6	869.9	978.8	88.8	0.1726	0.2048	0.1535
Colour ^4^																
L*	57.47 ^a^	50.93 ^b,c^	47.15 ^c,d^	44.06 ^d^	50.65 ^b,c^	50 ^b,c^	50.25 ^b,c^	48.59 ^b,c,d^	50.48 ^b,c^	50.63 ^b,c^	49.49 ^b,c,d^	53.43 ^a,b^	1.73	0.0018	0.8265	0.0002
a*	4.75	5.73	6.54	6.72	5.76	6.74	5.60	5.80	6.08	6.72	7.80	6.77	0.94	0.7647	0.0745	0.5778
b*	17.43 ^b,c^	20.45 ^a,b^	18.74 ^a,b,c^	20.35 ^a,b^	15.98 ^c^	19.48 ^a,b^	20.46 ^a,b^	19.57 ^a,b^	20.99 ^a,b^	20.12 ^a,b^	21.66 ^a^	20.73 ^a,b^	1.07	0.0390	0.0227	0.2827
Day 7																
pH	6.07	5.97	6.04	6.13	6.18	6.09	5.89	5.84	5.95	5.87	5.96	5.95	0.07	0.0968	0.5520	0.4604
Drip loss%	3.53 ^a^	3.27	3.33 ^a,b^	3.05 ^a,b^	2.18 ^b^	3.25 ^a,b^	2.65 ^a,b^	3.16 ^b^	2.38 ^a,b^	3.31 ^a,b^	3.38 ^ab^	3.5 ^a^	0.39	0.3103	0.0128	0.9328
Cooking loss%	25.45 ^a^	19.47	22.22 ^a,b^	23.81 ^a^	22.13 ^a,b^	23.43 ^a^	18.17 ^b^	18.83 ^b^	20.59 ^a,b^	24.04 ^a,b^	19.83 ^a,b^	22.63 ^a,b^	1.91	0.4316	0.0206	0.7182
Shear force (g)	1148.9	1152.1	1101.6	1137.3	1226.8	1008.2	945.2	1133.2	995.2	820.1	898.2	1067.9	105.3	0.3495	0.5628	0.8429
Color																
L*	55.01	47.79	51.22	52.82	52.95	51.52	50.25	50.24	50.76	51.43	50.33	50.89	1.51	0.2781	0.1628	0.4581
a*	6.28	6.73	7.19	6.20	7.15	7.09	7.00	7.58	7.58	6.57	7.14	6.57	0.69	0.9400	0.1152	0.8437
b*	20.61	18.11	20.79	20.83	20.29	21.67	22.21	21	21.25	20.18	19.92	19.43	0.81	0.1433	0.0571	0.7256

^a,b,c,d^ Means with different superscripts in the same row indicates significant difference (*p* < 0.05). ^1^ Dietary treatments: NC (negative control) = basal diet, PC (positive control) = basal diet + vitamin E (100 mg/kg feed), BS 0.25 = basal diet + 0.25% brown seaweed, BS 0.50 = basal diet + 0.50% brown seaweed, BS 0.75 = basal diet + 0.75% brown seaweed, BS 1 = basal diet + 1% brown seaweed, BS 1.25 = basal diet + 1.25% brown seaweed, GS 0.25 = basal diet + 0.25% green seaweed, GS 0.50 = basal diet + 0.50% green seaweed, GS 0.75 = basal diet + 0.75% green seaweed, GS 1 = basal diet + 1% green seaweed, GS 1.25 = basal diet + 1.25% green seaweed. ^2^ SEM = standard error of means. ^3^ Contrast *p*-Values = orthogonal polynomial contrasts of dietary increasing brown and green seaweed inclusion levels (0.0 to 1.25%). ^4^ Color: L* = lightness, a* = redness, b* = yellowness.

## Data Availability

Not applicable.

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
