# Peer review of "Brown and Green Seaweed Antioxidant Properties and Effects on Blood Plasma Antioxidant Enzyme Activities, Hepatic Antioxidant Genes Expression, Blood Plasma Lipid Profile, and Meat Quality in Broiler Chickens"

_animals, 2023, doi:10.3390/ani13101582_

Round 1

Reviewer 1 Report

The presented paper discusses an interesting issue related to Brown and Green Seaweed Antioxidant Properties and Effects on Blood Plasma Antioxidant Enzymes Activities, Hepatic Anthioxidant Genes Expression, Blood Plasma Lipid Profile and Meat Quality in Broiler Chickens. Issues related to seaweed and the possibility of using them as a feed additive or a component of drinking supplements for various animal species have been the subject of many studies for years. The presented research, in the aspect of searching for alternatives to the source of protein in the feed, as well as strong natural antioxidants that are necessary in the diet of animals, especially those related to food production on an industrial scale seems to be quite interesting.

Detailed comments (strong and poor sides of study/lack of information) :

- multidirectional analysis of the issue of oxidative stress, with no data on the expression of genes - oxidative stress markers in the blood, although such an analysis was performed in the liver. Determination of gene expression both in blood and in other tissues (apart from the liver, muscle tissues seem to be crucial for this type of determination in poultry production) would allow for a full analysis of the potential antioxidant activity within several tissues (lack of analysis the same genes in blood/other tissue)

-Methodology in terms of meat quality is correct

-the number of repetitions and the analysis of the effect of several doses gives wider possibilities of analysis and inference

Introduction -  need to be shorten and leave key information, in its current form it is too long,

Table 1.2 - is hard to read, ME values are merging, please either change the font or the way of writing (is it necessary to give decimal places?) Table - the same,

What specific type of algae was used in the study, please indicate the species name (e.g. Fuccus, Sargassum etc.), brown and brown seaweed is too general  statement. How were they obtained - were they ready-made, commercial raw materials? The data on the seaweed itself is too general and needs to be supplemented.

Were expression analyzes performed in the study for the same genes in the blood as in the liver? if yes, what were the results, if not why?

The discussion lacks information on how the obtained results may affect what is crucial for broiler chicken breeding and what is important from the breeder's and consumer's point of view (impact on physical and chemical parameters, etc.).

Was the meat amino acid profile analyzed in the study? Seaweed is considered a rich source of amino acids and such a parameter, in addition to the analysis of the lipid profile (also in meat), is worth analyzing.

I hope that the above comments and suggestions will improve the study and speed up its publication.

Author Response

Please find an attached file for the responses made according to the comments.

Reviewer 2 Report

This study was conducted to study the effects of brown and green seaweed on the antioxidant capacity, lipid profile and meat quality of broilers. I think it's an interesting study. However, there are some problems:

1.     The supplementation of VE has no obvious effect on the tested parameters, except for the L* value at day1. Thus, can VE be used as positive control in this study?

2.     Line 59: Although the full names of BS, GS, CP were given in the abstract, the full names should also be given at the first time in the main text.

3.     Lines 65-69: What does ng1/mg mean? Should it be ng/mg?

4.     Line174: To what extent was the seaweed sample crushed? Was the seaweed sample passed some mesh sieves?

5.     Line179: Were the blood centrifuged at 35000g, such a high speed? Usually, plasma was separated at ~2000 g (doi.org/10.3382/ps/pex395). Please check.

6.     Gene names in Tables and text should be in italic

7.     Line284: Should it be 1.5 mL tube?

8.     Line 356: Add ‘p < 0.05’.

9.     Line365: The authors write ‘GPx enzyme activity’, but according to Table 4, the unit for Gpx is μM/L. What does the μM/L mean? I think it is a content unit.

10.  Line402: This sentence should be more accurate, as compared to the PC group, the TC and HDL levels in the BS1 group were not significantly higher than that in the PC group.

11.  Table 8: l*should be L*, as consistent with the 2.8.2.

The quality of English is satisfactory.

Author Response

(The authors gave the same response as above.)

Round 2

Reviewer 1 Report

Dear Authors,

Thank you for the revised manuscript. The tables are still difficult to read but I understand that it is impossible to make them more visible and friendly for the reader and you do not want to put them as supplementary material (you can put them in a different format).

The argument related to the genes and liver-is to general. There many studies where the same genes- also related to the oxi-redox status- where checked in a different tissues, organs etc. after using feed additives/antibiotics etc. and showed different expression which was mostly related to the final concentration of the active compounds, pharmacokinetics etc.  However,   I understand that such analyzes are not intended for this study.

Please, use the reference format as is recommended for Animals MDPI (e.g. journal abbreviations).

Reviewer 2 Report

The authors have revised the problems according to the comments. I think this manuscript can be accepted.
